# Assessing Soil Erosion by Monitoring Hilly Lakes Silting

Yamuna Giambastiani [1,*], Riccardo Giusti [1], Lorenzo Gardin [1], Stefano Cecchi [1], Maurizio Iannuccilli [1], Stefano Romanelli [2], Lorenzo Bottai [2], Alberto Ortolani [1,2] and Bernardo Gozzini [1,2]

1    CNR-IBE, National Research Council, Institute of Bioeconomy, 50019 Florence, Italy; giusti@lamma.toscana.it (R.G.); gardin@lamma.toscana.it (L.G.); cecchi@lamma.toscana.it (S.C.); iannuccilli@lamma.toscana.it (M.I.); ortolani@lamma.toscana.it (A.O.); gozzini@lamma.toscana.it (B.G.)
2    Environmental Modelling and Monitoring Laboratory for Sustainable Development, LaMMA Consortium, 50019 Florence, Italy; romanelli@lamma.toscana.it (S.R.); bottai@lamma.toscana.it (L.B.)
*    Correspondence: giambastiani@lamma.toscana.it

**Abstract:** Soil erosion continues to be a threat to soil quality, impacting crop production and ecosystem services delivery. The quantitative assessment of soil erosion, both by water and by wind, is mostly carried out by modeling the phenomenon via remote sensing approaches. Several empirical and process-based physical models are used for erosion estimation worldwide, including USLE (or RUSLE), MMF, WEPP, PESERA, SWAT, etc. Furthermore, the amount of sediment produced by erosion phenomena is obtained by direct measurements carried out in experimental sites. Data collection for this purpose is very complex and expensive; in fact, we have few cases of measures distributed at the basin scale to monitor this phenomenon. In this work, we propose a methodology based on an expeditious way to monitor the volume of hilly lakes with GPS, sonar sensor and aquatic drone. The volume is obtained by means of an automatic GIS procedure based on the measurements of lake depth and surface area. Hilly lakes can be considered as sediment containers. Time-lapse measurements make it possible to estimate the silting rate of the lake. The volume of 12 hilly lakes in Tuscany was measured in 2010 and 2018, and the results in terms of silting rate were compared with the estimates of soil loss obtained by RUSLE and MMF. The analyses show that all the lakes measured are subject to silting phenomena. The sediment estimated by the measurements corresponds well to the amount of soil loss estimated with the models used. The relationships found are significant and promising for a distributed application of the methodology, which allows rapid estimation of erosion phenomena. Substantial differences in the proposed comparison (mainly found in two cases) can be justified by particular conditions found on site, which are difficult to predict from the models. The proposed approach allows for a monitoring of basin-scale erosion, which can be extended to larger domains which have hilly lakes, such as, for example, the Tuscany region, where there are more than 10,000 lakes.

**Keywords:** sediment monitoring; remote sensing; lakes; water capacity; sonar; aquatic drone; USLE; MMF




## 1. Introduction

### 1.1. Erosion: A Worldwide Threat

After almost a century of research and studies on the territory, soil erosion caused by water, wind and tillage is known to be the greatest threat to soil health, and to the ecosystem services it provides, in many regions of the world [1–3]. Its impact on global crop production has been estimated at a reduction of 0.4% per year [4]. Some authors argue that nearly a third of the world's arable land has been lost due to erosion over the past 40 years and continues to decrease at a rate greater than ten million hectares per year [5]. Erosion is a natural phenomenon which consists of the loss of the most superficial layer of the soil due to the action of precipitation or wind. With the advent of modern agriculture and, above all, with (I) the introduction of extensive mechanization, (II) the leveling of the

slopes, (III) the abandonment of traditional hydraulic-agricultural solutions and (IV) the specialization of crops, erosion has assumed worrying proportions [6–8]. Erosion is now a worldwide threat, especially in hilly areas with significant economic impacts, particularly in areas with valuable crops [9,10]. Water erosion represents one of the main threats to the correct functionality of the soil, through (I) the removal of the fertile surface soil horizon, (II) the denser subsoil incorporation in the surface layer and (III) the possible decrease in the root zone [11].

A reliable assessment of this phenomenon is therefore particularly useful as a decision-support tool for planning soil conservation interventions [12–14]. To address these issues, the Community Agricultural Policy has ensured that agriculture is in line with the EU soil protection policies. Effective management of these issues is considered essential for many strategies and priorities of the European Green Deal, as defined primarily in the (I) thematic strategy and the sustainable management of soil [15], (II) the fight against erosion, and (III) the fight against the loss of organic carbon and biodiversity in soil. The quantitative assessment of soil erosion, due to both water and wind, is generally carried out through modeling the phenomenon or with experimental tests (plots, rain simulators, etc.) carried out directly on the field [16]. In recent decades, researchers in Italy have also conducted several direct studies on the phenomenon of erosion [17–22].

### 1.2. Models for Erosion Estimation

The most commonly used erosion estimation model is the universal equation of soil loss (USLE) [23], and its revised version (RUSLE) [24], which estimates the annual mean long-term loss of soil due to sheet (interrill) and rill erosion. It should be noted that soil loss caused by (ephemeral) gully erosion is not predicted by RUSLE [25]. Despite its shortcomings, RUSLE is still the most widely used model on a large scale [26,27]. It can process data input for large regions and provides a basis for scenario analysis and taking actions against erosion [28]. A recent work [29] estimated erosion on a European scale using the most in-depth processing of the single factors [29–32]. USLE has been applied in comparative studies between various analysis methods, and the authors have shown that it does not lead to greater errors than process-based physical models (WEPP and PESERA), although it has some limitations due to the simple empirical nature of the model [33–35]. Soil loss is also analyzed worldwide through the revised MMF—Morgan–Morgan–Finney model [36,37], in order to evaluate the land degradation and ecological status of specific catchment areas or wider territories [38–40]. This model allows an erosion simulation to be developed in relation to the characteristics of the vegetation cover. The comparison between these empirical models shows similar results [41,42]. The Soil and Water Assessment Tool (SWAT) is a semi-empirical model used for the assessment of erosion phenomena at the basin scale [43–45]. It also allows analyses related to hydrological processes [46], land management and climate change [47,48]. Other simulations have been performed directly on reconstructions of hydrographic basins in miniature or in experimental sites [49–51].

In Tuscany in 2009, soil erosion estimates were made at a regional scale with the USLE model through the elaboration of single climatic, pedological, land use and morphometric factors at high resolution [52,53]. Direct measurements carried out over the years in experimental fields [54–56] have allowed the model to be properly constrained and tested.

### 1.3. Scope of Work

With this work, we propose a new approach to monitor erosion phenomena at the basin scale, based on an expeditious estimate of the hilly lakes silting rate, through remote sensing techniques. The basic assumption is that a relationship exists between the soil loss (or sediment production) from the basin with the volume loss of the reservoir. The silting rate, estimated by sonar and aquatic drone [57], is compared with the soil loss obtained from two models (RUSLE and MMF) in order to evaluate erosion through direct measurements of the sediment produced. The hilly lakes distributed throughout the territory can be considered the containers of the sediment coming from the erosion phenomena of the

slope, and therefore constitute a net of distributed monitoring. This expeditious method of estimating the reservoir capacity for the study of erosion phenomena is an innovative tool that enables the estimation of the sediment produced by a slope. The aim is to demonstrate that through the repetition over time of a simple procedure of silting estimation, it is possible to observe the evolution of the erosive phenomena and better understand the impact of anthropogenic actions or climate change on the quality of soils. In Tuscany, there are about 5000 lakes with a surface greater than 1000 square meters, which can become the mean for the distributed monitoring of erosive phenomena.

## 2. Materials and Methods

### 2.1. Lakes Analyzed and Volume Changes

The study took into consideration 12 hilly lakes in Tuscany, shown in Figure 1, of which the volume calculated in 2010 is known, thanks to a past survey by the former Agency for Development and Innovation in Agriculture (ARSIA) of Regione Toscana (RT—regional administration of Tuscany). The volume estimate was carried out by a private company (Aquaterra, Florence) using a boat, a depth sounder and GPS, thus carrying out a bathymetric survey [58]. LaMMA Consortium, applying the methodology described by Giambastiani et al. 2020 [57], measured the lakes again in 2018 thanks to a monitoring project. We assume that the two methodologies are comparable, as the same types of tools and procedures are used. Furthermore, the measurements were carried out for both years in early spring, when the reservoir tends to be full from winter rains and agricultural use is limited. Each lake and its basin was evaluated and investigated in order to carry out a modeling analysis of the surface erosion with the common models (RUSLE, MMF). These lakes are mainly used for irrigation of agricultural crops; however, some are used for sport fishing, forest-fire-fighting and other objectives. Table 1 shows the main characteristics of the basins corresponding to such lakes. Table 2 reports historical data regarding the lakes volume at the time of construction. The comparison between 2010 and 2018 is summarized in Table 3.

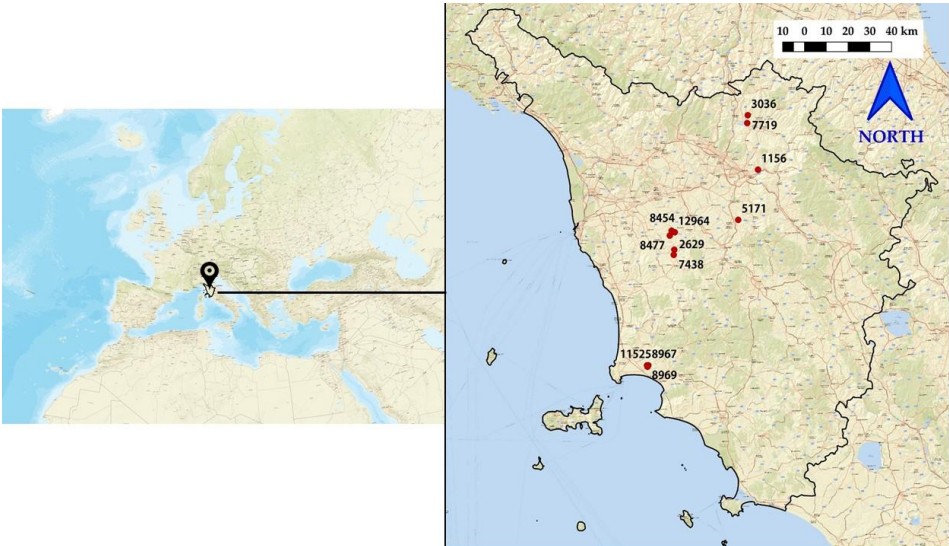

**Figure 1.** Lakes geographic distribution in Tuscany, Italy.

In order to implement the erosion estimation models (RUSLE and MMF), data for the hydrographic basins were collected relative to the precipitation (Figure 2) of the meteorological stations closest to the lakes; the hydrological network was elaborated, for each basin, from a DTM (Digital Terrain Model) with resolution 10 × 10 m (Figure 3); and land cover was processed via photointerpretation (Figure 4, Table 4) in 9 main classes.

**Table 1.** Main characteristics of the basins corresponding to the studied lakes (Appendix A): Altitude and slope are obtained from a DTM with 10 m resolution. Hydrographic networks are taken from a database of the regional administration of Tuscany (https://www.regione.toscana.it/-/geoscopio, accessed on 1 April 2018); viability is taken from the OpenStreetMap database.

| GID | Lake Name | Area (ha) | Altitude Max (m agl) | Altitude Lake (m agl) | Slope Mean (%) | Hydrographic Network (m) | Road Network (m) |
|---|---|---|---|---|---|---|---|
| 1156 | Romena | 9.55 | 293 | 154 | 15.1 | 618.8 | 331.5 |
| 2629 | Cavalcanti | 61.64 | 212 | 156 | 10.9 | 2594.6 | 751.4 |
| 3036 | Galliano | 67.25 | 409 | 281 | 8.3 | 3515.7 | 2883.1 |
| 5171 | Fabbrica | 218.99 | 413 | 229 | 14.1 | 11,268.2 | 11,812.8 |
| 7438 | Pavone | 50.83 | 202 | 135 | 14.4 | 1998.9 | 1029.9 |
| 7719 | Schifanoia | 87.04 | 281 | 242 | 4.6 | 4549.5 | 2297.4 |
| 8454 | Castelfalfi 1 | 52.70 | 261 | 158 | 14.8 | 2459.3 | 4034.4 |
| 8477 | Castelfalfi 3 | 127.19 | 177 | 99 | 10.3 | 7852.2 | 6069.4 |
| 8967 | Potenti 2 | 65.34 | 183 | 48 | 11.5 | 4135.1 | 1091.1 |
| 8969 | Potenti 1 | 43.79 | 128 | 39 | 9.4 | 2351.5 | 0.0 |
| 11525 | Angiola | 136.16 | 195 | 35 | 14.6 | 8183.8 | 11,074.5 |
| 12964 | Castelfalfi 2 | 64.00 | 338 | 177 | 17.5 | 3507.2 | 1476.0 |

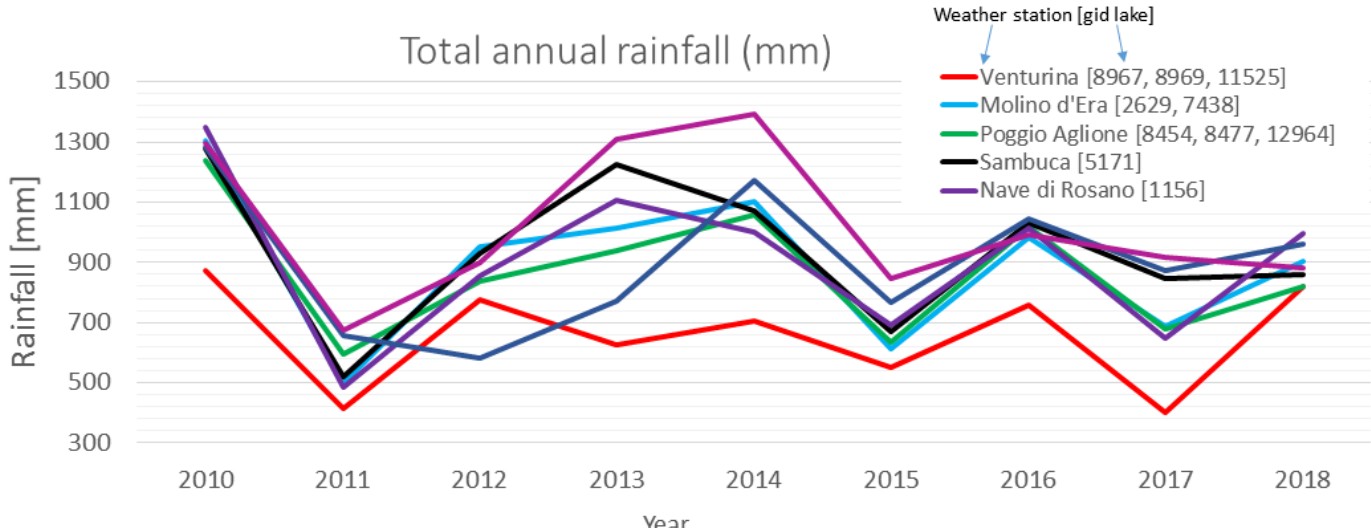

**Figure 2.** Rainfall trends in the period object of study, for the nearest weather stations. The legend on the right side associates each weather station to the corresponding lake(s).

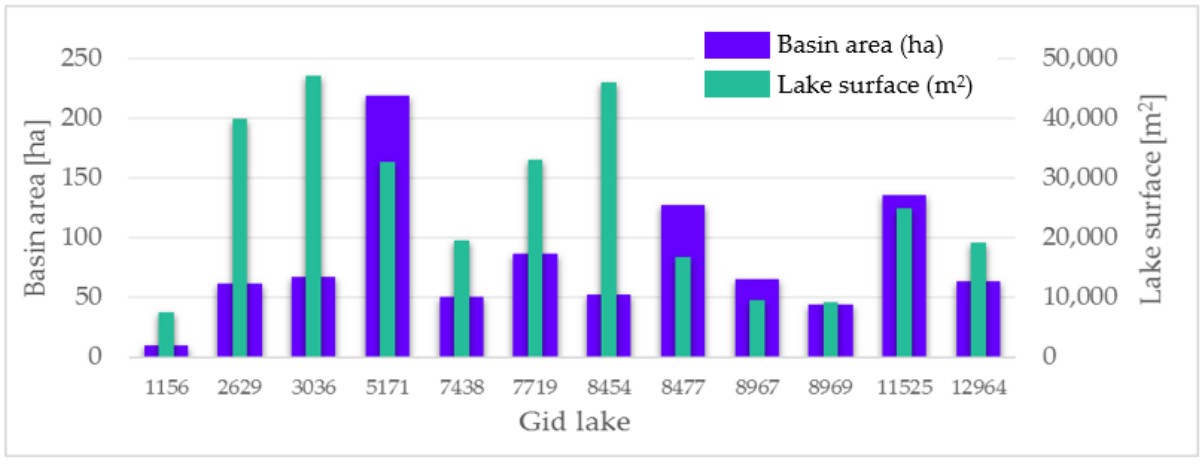

**Figure 3.** Relationships between lake surface and corresponding drainage basin.

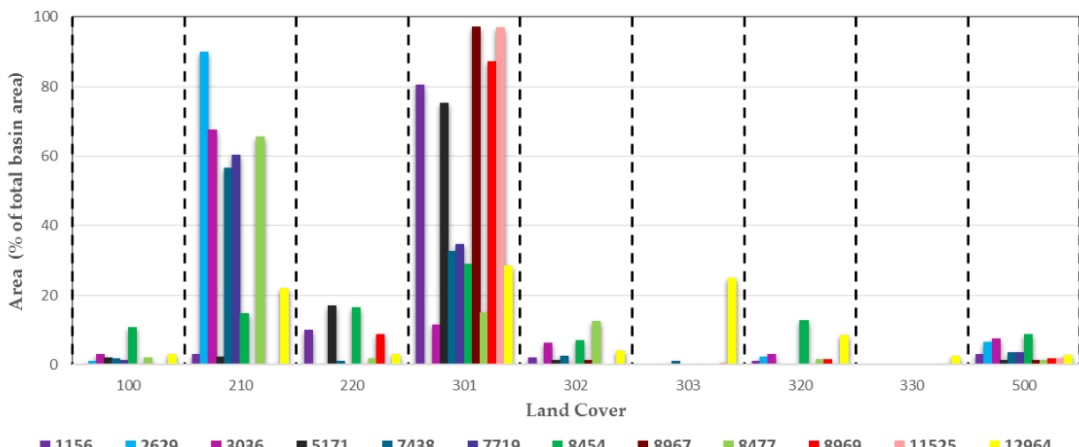

**Figure 4.** Distribution of land-cover classes for each lake. The land-use classes are shown along the x axis, described in Table 4, while the legend shows the lake GID. In X axis: 100 = artificial surfaces; 210 = lands under a rotation system used for annually harvested plants and fallow lands; 220 = permanent crops (vineyards and olive groves); 301 = forest with a complete canopy closure or a little less; 302 = forest with a sparse canopy closure (40–60%), shrubs and max 10% of soil bare; 303 = degraded forest (canopy closure less of 40%), shrubs cover of 40% and bare soil max 30%; 320 = permanent shrub and/or herbaceous vegetation associations; 330 = degraded soil or bare rock; 500 = water bodies.

From research conducted in the archives of the body in charge of the regulation and authorization of artificial lakes, it was also possible to recover the lake volume on the project deposited during the authorizations phase. From what has been learned, however, unregistered changes have often occurred regarding the morphology of the reservoir, so the related dimensional parameters are to be considered just indicative (Table 2, Figure 5).

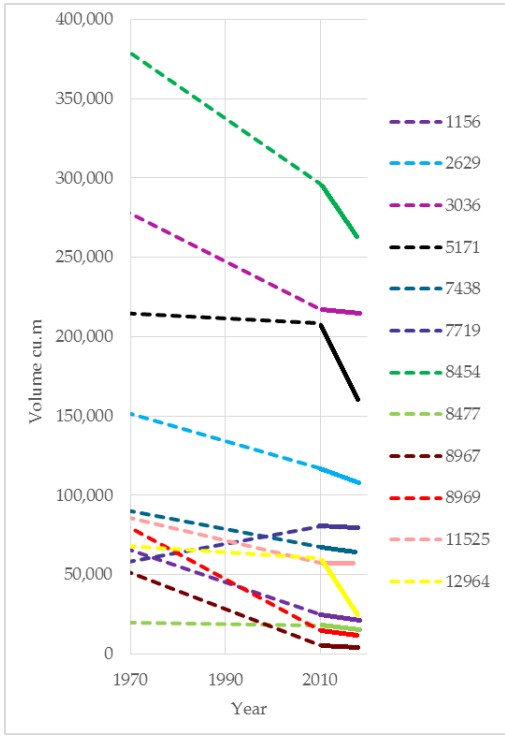

**Figure 5.** Trend of the reservoir capacity from construction to the analysis period. Dashed lines indicate the uncertain range, while solid lines indicate the trend found with the actual analysis.

**Table 2.** Construction year and project volumes.

| GID | Construction Year | Volume of Design Phase (mc) |
|---|---|---|
| 1156 | 1964 | 76,000 |
| 2629 | 1958 | 160,000 |
| 3036 | 1958 | 293,150 |
| 5171 | 1956 | 216,420 |
| 7438 | 1959 | 96,000 |
| 7719 | 1970 | 52,500 |
| 8454 | 1970 | 400,000 |
| 8477 | 1963 | 20,507 |
| 8967 | 1970 | 63,000 |
| 8969 | 1970 | 96,000 |
| 11525 | 1970 | 93,000 |
| 12964 | 1967 | 69,803 |

**Table 3.** Lake parameters for the years 2010 and 2018: surface area, volume, variation (in volume, percentage and percentage per year), and silting rate, the latter normalized to the lake surface area. For 2010, we show harmonized volumes, indicated as 2010 h.

| GID | Surface (m$^2$) | | Volume (m$^3$) | | Variation | | | Silting | |
|---|---|---|---|---|---|---|---|---|---|
| | 2010 | 2018 | 2010-h | 2018 | m$^3$ | % | %/y | Mg | Mg/y |
| 1156 | 7570 | 7599 | 24,855 | 21,597 | −3258 | −13.1 | −1.6 | 2821 | 353 |
| 2629 | 38,875 | 39,942 | 116,826 | 108,254 | −8572 | −7.3 | −0.9 | 7423 | 928 |
| 3036 | 49,986 | 47,241 | 217,520 | 215,144 | −2376 | −1.1 | −0.1 | 2057 | 257 |
| 5171 | 35,293 | 32,713 | 208,807 | 159,454 | −49,353 | −23.6 | −3.0 | 42,740 | 5342 |
| 7438 | 20,729 | 19,561 | 67,659 | 64,228 | −3431 | −5.1 | −0.6 | 2971 | 371 |
| 7719 | 35,080 | 33,029 | 80,651 | 79,407 | −1244 | −1.5 | −0.2 | 1077 | 135 |
| 8454 | 48,412 | 46,044 | 296,296 | 261,833 | −34,463 | −11.6 | −1.5 | 29,845 | 3731 |
| 8477 | 13,389 | 16,747 | 18,018 | 15,315 | −2703 | −15 | −1.9 | 2341 | 293 |
| 8967 | 8059 | 9654 | 5792 | 4141 | −1651 | −28.5 | −3.6 | 1430 | 179 |
| 8969 | 7744 | 9180 | 15,220 | 12,070 | −3150 | −20.7 | −2.6 | 2728 | 341 |
| 11525 | 21,246 | 24,886 | 57,625 | 57,238 | −387 | −0.7 | −0.1 | 335 | 42 |
| 12964 | 22,135 | 19,204 | 60,549 | 23,953 | −36,596 | −60.4 | −7.6 | 31,692 | 3961 |

**Table 4.** Land cover classes corresponding to C and P factors.

| UCS Code | Description | USLE_C | USLE_P |
|---|---|---|---|
| 100 | Artificial surfaces | 0 | 1 |
| 210 | Lands under a rotation system used for annually harvested plants and fallow lands | 0.15 | 1 |
| 220 | Permanent crops (vineyards and olive groves) | 0.4 | 1 |
| 301 | Forest with a complete canopy closure or a little less | 0.01 | 1 |
| 302 | Forest with a sparse canopy closure (40–60%), shrubs and max 10% of soil bare | 0.08 | 1 |
| 303 | Degraded forest (canopy closure less of 40%), shrubs cover of 40% and bare soil max 30% | 0.20 | 1 |
| 320 | Permanent shrub and/or herbaceous vegetation associations | 0.1 | 1 |
| 330 | Degraded soil or bare rock | 0.75 | 1 |
| 500 | Water bodies | 0 | 1 |

Harmonization

In order to compare the two volume measures, a lake-surface-based harmonization was applied, as this parameter (surface area) was easily obtained from the Tuscany Region orthophotos at 20 cm resolution (https://www.regione.toscana.it/-/geoscopio, accessed on 1 April 2018). Harmonization is necessary because in two different years we could have a different reservoir capacity due to the water level, according to different previous rainfall. It is based on the surface variation between 2010 and 2018 (Figure 6), according to the following equations.

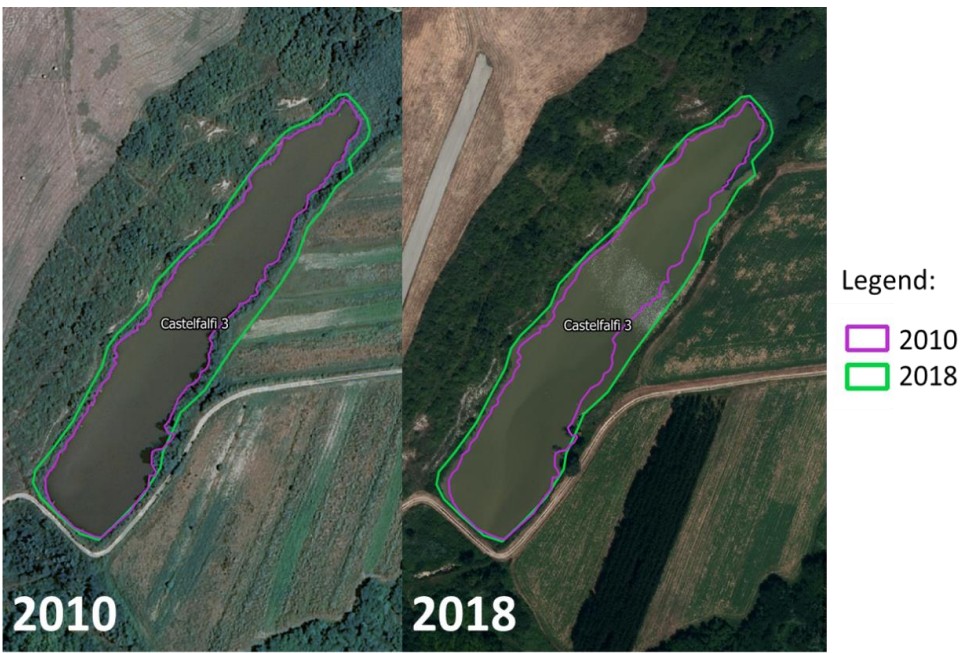

**Figure 6.** Orthophotos of lake "gid 8477" where it is possible to check the difference in water level.

We can write a generic expression for the lake volume *V*, according to Giambastiani [57], as:

$$V = \iint h(x,y) \cdot dx dy = \int_S h \cdot d\sigma \tag{1}$$

where *h* is the height that depends on the coordinate positions (*x*,*y*), which we have omitted in the second step, rewriting the integral as a surface one ($\sigma$). Using the integral mean value theorem, we can write *V* as:

$$V = \langle h \rangle_S \cdot S \tag{2}$$

$\langle h \rangle_S$ being the lake height average (over the surface *S*). If we assume that its variation is negligible for limited variation of *S*, we can write the variation $\Delta V$ of the volume as a linear function of $\Delta S$, the latter being the variation of the surface with time (depending for instance to rainfall, evaporation, etc.).

$$\Delta V = \langle h \rangle_S \cdot \Delta S = \underline{h} \cdot \Delta S \tag{3}$$

The last step is just to rename $\langle h \rangle_S$ with $\underline{h}$, both for simplicity and to highlight the assumption that it is no more dependent on the surface extension *S*.

In practice, we have measured $\underline{h}$ for the reference year $y_0$, which for us was the year 2010, for which we had the ARSIA measurements of the surface areas with the corresponding volumes.

Where the "Surface measured 2010" is the ARSIA surface, measured simultaneously with the volume

$$\underline{h} = \frac{V_{y_0}}{S_{y_0}} \tag{4}$$

For a generic year *y*, the volume to be compared with the one at the reference year $y_0$ becomes:

$$V_y = V_{y_0} + \Delta V = V_{y_0} + \underline{h} \cdot \Delta S \tag{5}$$

with $\Delta S$ as the surface difference between 2018 and 2010; surfaces were obtained through photointerpretation of orthophotos.

*2.2. Erosion Simulation by the Morgan–Morgan–Finney Model*

The MMF model divides the soil erosion process into two phases: the phase related to the water component, which determines the energy of the rainfall, and the phase related to the production of sediments, based on the characteristics of the soil. Soil loss in relation to erosion is determined on the basis of precipitation and transport capacity, influenced by soil cover and slope [41,59]. The MMF model is implemented within the open-source SAGA GIS software. Input data come from various sources. Starting from the Digital Terrain Model (DTM), with a resolution of 10 m, the slope map and the channel network were elaborated, while the "plant height" map was obtained through the Crown Height Model (CHM—10 × 10 m). Canopy cover, permanent interception and ground cover derive from Sentinel-2 image processing, in particular based on the NDVI calculation [60], with 10 m resolution. The characteristics of the soils (bulk density, effective hydrological depth, percentages of clay, sand and silt, etc.) are derived from the soil database of RT (http://www502.regione.toscana.it/geoscope/pedologia.html, accessed on 1 April 2018). Other necessary input variables for the model were obtained from direct processing of the land use and land cover map, carried out by photointerpretation. Annual precipitation data were taken from the meteorological stations closest to the lakes in question. In particular, the average distance between the lakes and the rain gauges is 4.5 km, with a standard deviation of 1.8 km.

*2.3. Erosion Estimate by RUSLE Model*

The Universal Soil Loss Equation (USLE) [23] and subsequent revisions (RUSLE) [24], is an empirical relationship, as it derives from experimental plots carried out in the United States and from the mathematical definition of the results found from these plots, which models soil erosion as a process resulting from a set of six main factors: the energy and intensity of precipitation (R factor), the erodibility of the soil (K factor), the length and slope of the plot (LS factor), vegetation cover (C factor) and conservation practices (P factor). The employed R factor is derived from the SIAS project (ISPRA 2016), through which a simplified relationship is elaborated between the amount of rainfall and the erosion value [61]. This simplification does not critically affect the accuracy, as it emerges from comparison work [31]. In fact, the mean value of R factor for Tuscany is 1765 (standard deviation = 710) against 1748 (standard deviation 365), as found by Panagos 2015 [29]. Larger fieldwork, developed in 2006 for punctual soil data, allowed the calculation of K factor carried out following the original methodology [23]. The LS factor was calculated for the basins of each lake, using a digital elevation model (DEM) of 10 × 10 m, with the method developed by Desmet and Govers 1996 [62]. In order to avoid overestimation of the LS factor in heterogeneous landscapes, the lengths of long slopes were limited to a value of 333 m [24,62]. The length exponent (m) is based on the original USLE method [63]. The C factor was completely updated, carrying out a detailed photo interpretation on orthophotos, using functional classes for the purpose. For each soil cover class identified, the values C and P were attributed as in Table 4, consistent with the values reported in the bibliography [24,31,64,65].

From the photo interpretation, it was possible to verify that in the agricultural landscapes of the study areas, there were no particular conservation techniques similar to those already codified in the RUSLE model: for this reason, we decided to always adopt a factor P = 1.

Sediment Delivery Ratio

The Sediment delivery ratio (SDR) is applied to estimate the amount of sediments produced by the erosion phenomena that reaches the lake [66]. SDR is the erosion fraction, generated by each single source cell, which reaches the nearest permanent drainage line. As a first approximation, the SDR can be considered constant for the whole basin or sub-basin [67], but in recent works, it is calculated pixel by pixel as a function of the length and slope of the path in the downstream direction [68]. The most complete al-

gorithm for its modeling is proposed by [65], which accounts for the connectivity index (IC) for each pixel, considering the morphological and hydrological characteristics of both the hydrological upstream and downstream portion of the pixel. It describes the hydrological link between sediment sources and collection and transfer points such as streams [69]. For SDR calculation in the study area, we have used the InVEST model (https://naturalcapitalproject.stanford.edu/, accessed on 3 September 2021), which implemented the algorithm of Borselli [65], making some minor simplifications.

## 3. Results

The silting rate (SR) (Figure 7) is very variable among the lakes under study and no significant relationships appear between the lake volume or the surface of the basin, or other main dimensional parameters. Some lakes have a high volume variation (e.g., 12964, 5171) and this is not related to either the size of the lake or the basin. The annual silting rate is obtained by dividing the volume variation by 8. We consider this operation significant as the surveys were carried out, in both cases, during the months of March and April, with a difference of a few weeks. The lakes' surfaces did not vary much during the 8 years of analysis. Figure 8a shows the relationship between the area of 2010 and 2018. The strong correlation indicates that the variation in the surface responds to ordinary dynamics. The surface change is present in almost all lakes; from this, we can confirm the need to harmonize volumes. While this process may be the source of further processing errors, we believe it is robust as the volume variations are small. A greater variation may exist in the relationship between the lakes' mean depths (Figure 8b). For all cases, we found that the mean depth has decreased. This is further confirmation of the soundness of the harmonization process. The correlation matrix highlights significant relationships between several parameters considered (Figure 9). Among these, we find a good correlation between soil loss and the erodibility of the lithology (K_lito_Sl, r = −0.72), the quantity of specialized (olive grove, vineyard) agricultural land cover (r = 0.85) and with the presence of roads (r = 0.68).

In Figure 9, we summarize the many relationships developed between the silting and the characteristics of the lake or basin. Among these are the physical characteristics of the basin, such as the altimetry and the difference in height (alti-max and altit_lake, disl_basin), the length of the network present in the basin (hydro_network), the presence of roads (road_net), the various soil classes (sup_ucs: Table 4) and rainfall (rain_acc, num_event).

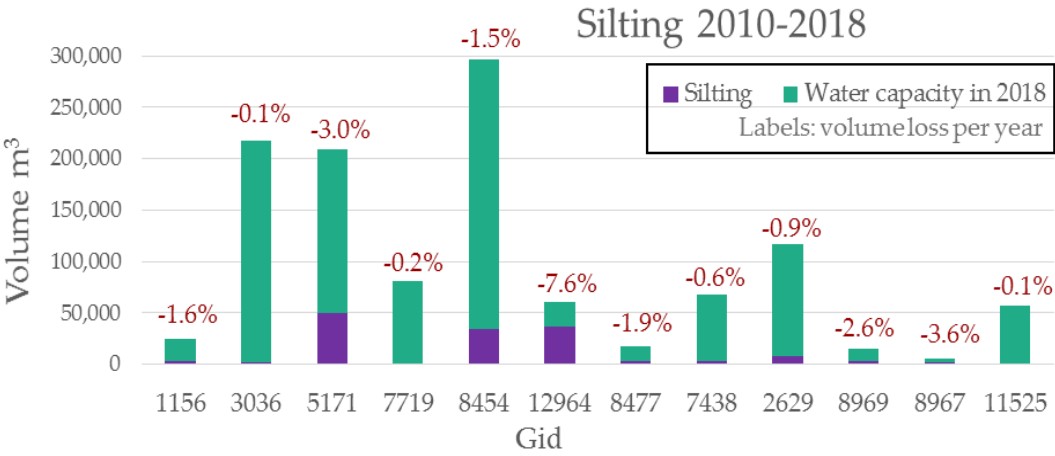

**Figure 7.** Silting rate and water capacity of the lakes under study.

Comparing the silting rate (SR) of each lake, obtained with the proposed methodology [57], with the annual soil-loss values obtained from the described empirical models, the link between these methodologies is evident (Figure 10). In absolute terms, the sediment values produced (SL) were almost always lower than the SR values. SL by RUSLE was similar to SR for three lakes. In seven cases, however, it was different, even if it was

consistent with the MMF results. For example, Lake 5171 had the highest silting rate with 3701 Mg y$^{-1}$, which was similar to the MMF estimate (3384), while with RUSLE, we found a third of the soil was lost. It should be noted that this lake has a much larger basin than all the others. Lake 8477 showed an inverse and very different trend compared to the models: 202 (SR), 630 (SL-RUSLE), 675 (SL-MMF). For other lakes, we found a close similarity. For example, lake 2629 had SR = 643; RUSLE = 499; MMF = 558 Mg y$^{-1}$. Additionally, in statistical terms, SR showed higher absolute values (SR/ha mean = 15.1; SL MMF/ha mean = 7.9; SL RUSLE/ha mean = 6.3).

Using the RUSLE and SAGA MMF models, it was possible to estimate the soil-loss rate of each catchment area [70], which was compared with the average silting rate of the reservoirs (considered as the closure section of the basin), in the period 2010–2018. The average sediment produced per hectare for the entire analysis sample was in line with the work of Angeli et al. 2004 [71]. The average annual soil loss per hectare obtained by RUSLE was 6.35 Mg, while MMF returned 7.96 Mg. The average silting per hectare of catchment area was 15.13 Mg y$^{-1}$ (Table 5). Both models showed a good correlation with the silting rate measured for each lake (Figure 11), but with a stronger relationship with the RUSLE model. The good significance of such relationships suggests a close link between the loss of soil and the silting rate, as visible in Figure 10.

**Table 5.** Summary of sediment volumes and average values per hectare, obtained from field surveys (silting) and models (soil loss for RUSLE and MMF).

| GID Lake | Basin Area (ha) | Silting (Mg/y) | SL RUSLE (Mg/y) | SL MMF (Mg/y) | Silting (Mg y$^{-1}$ ha$^{-1}$) | RUSLE (Mg y$^{-1}$ ha$^{-1}$) | MMF (Mg y$^{-1}$ ha$^{-1}$) |
|---|---|---|---|---|---|---|---|
| 1156 | 10 | 244 | 49 | 4 | 24.393 | 4.928 | 0.422 |
| 2629 | 60 | 643 | 499 | 559 | 10.703 | 8.306 | 9.299 |
| 3036 | 81 | 178 | 344 | 609 | 2.197 | 4.242 | 7.511 |
| 5171 | 213 | 3701 | 1248 | 3384 | 17.390 | 5.862 | 15.900 |
| 7438 | 50 | 257 | 290 | 871 | 5.183 | 5.842 | 17.543 |
| 7719 | 101 | 93 | 192 | 7 | 0.925 | 1.899 | 0.071 |
| 8454 | 41 | 2585 | 655 | 721 | 63.529 | 16.092 | 17.733 |
| 8477 | 106 | 203 | 630 | 675 | 1.906 | 5.925 | 6.349 |
| 8967 | 53 | 124 | 11 | 3 | 2.341 | 0.209 | 0.049 |
| 8969 | 32 | 236 | 17 | 3 | 7.329 | 0.523 | 0.088 |
| 11525 | 135 | 29 | 31 | 1 | 0.215 | 0.229 | 0.004 |
| 12964 | 60 | 2745 | 1342 | 1247 | 45.475 | 22.231 | 20.653 |

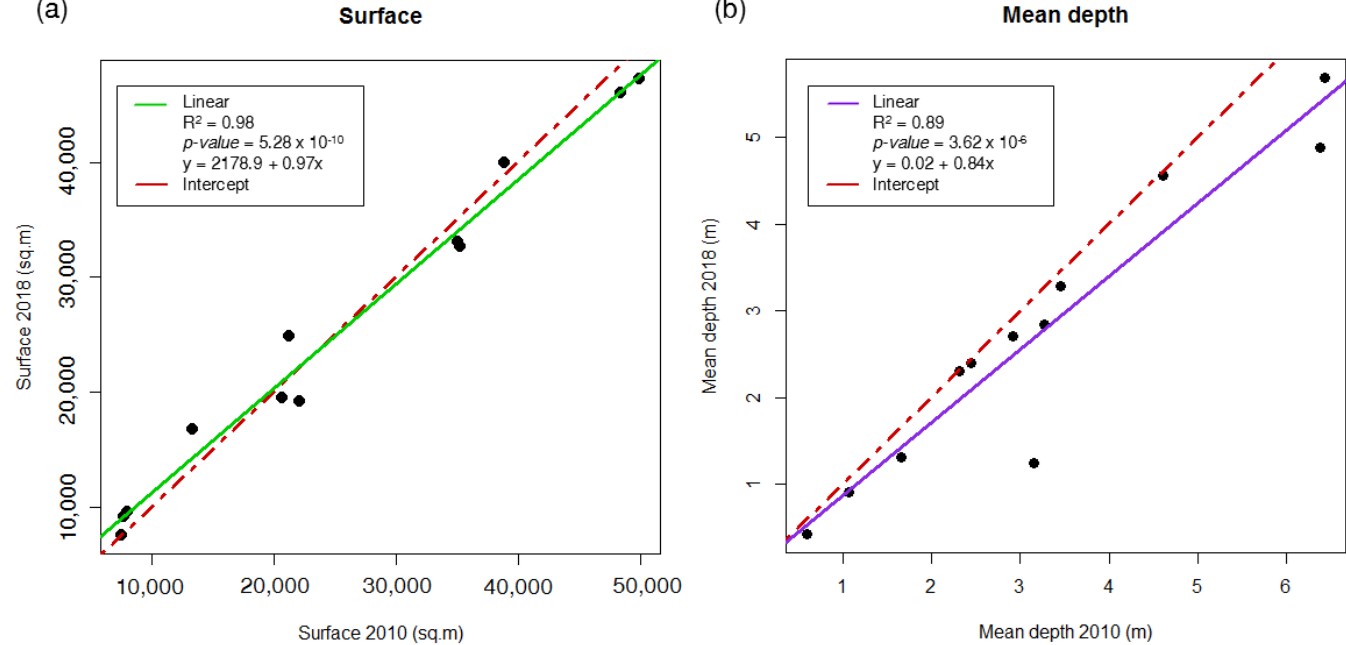

**Figure 8.** Relationships between variations in surface area (**a**) and mean depth (**b**).

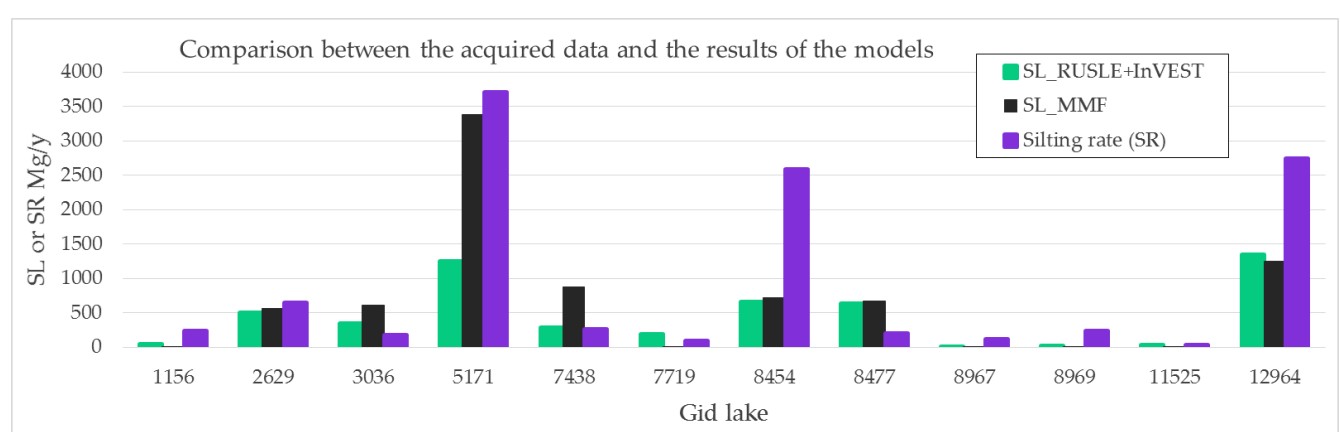

**Figure 9.** Correlation matrix, based on Pearson's analysis.

**Figure 10.** Comparison between the soil loss (SL) calculated with the models (RUSLE and MMF) and the silting rate for each individual lake.

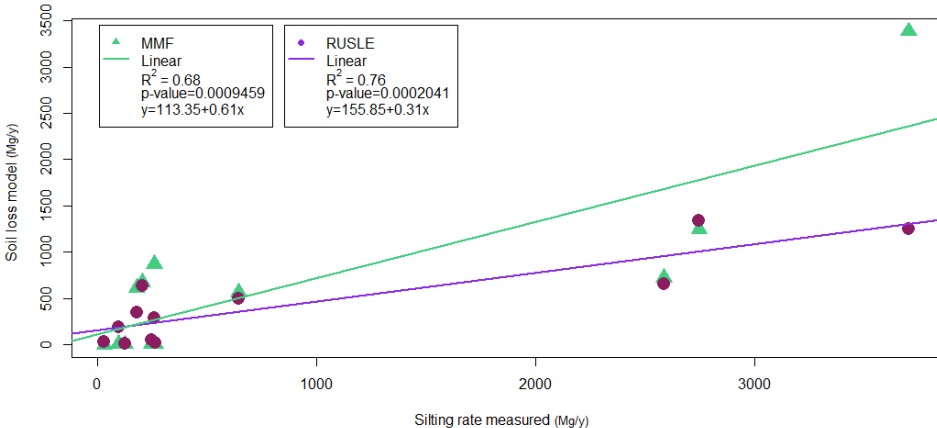

**Figure 11.** Relationship between the measured silting rate and the soil loss obtained from the MMF and RUSLE models.

## 4. Discussion

The methods that allow the monitoring of erosive phenomena are often very expensive to apply: the models require several input variables and therefore (I) challenging surveying campaigns, (II) data collection from different sources to be harmonized and (III) aerial images and high-resolution digital terrain models for GIS analysis and photointerpretation. All this involves high costs, low reaction time in carrying out analyses at critical moments, and possible relevant evaluation errors due to the high number of interactions and variables to take into account [4]. The proposed approach for the erosion evaluation is based on monitoring the water capacity of hilly lakes, which are considered as real sediment containers. The variation in lake volume, following silting, can be linked to the production of sediments from the afferent basin. Given the high distribution of artificial hilly reservoirs in the Tuscan territory, the proposed approach could create conditions for a periodic assessment of the degradation phenomena of the soil and the territory in general. The use of input data with greater accuracy and precision, such as using LiDAR Dems [70] and an in situ meteorological station for precipitation measurements, could lead to improvements. This approach is also easily applicable to different contexts, as long as they have a good distribution of hilly reservoirs. The methodology, however, has some limitations for calculating the volume of the lake, in particular due to the simplification of the shape of the lake profile [57]. Anyway, the application of this methodology, from a monitoring point of view, can be valuable for reducing major evaluation errors arising from too-long sampling-time intervals that we have when using classical approaches. In fact, field measurements for medium-sized reservoirs (up to 4–5 hectares of surface area) can be carried out in a short time (e.g., 30 min), and processing can be carried out automatically within a few minutes. This allows a wide and rapid application, analyzing wide domains with relatively low effort and costs. The results obtained show significant measures of the lake volume variation, well correlated with other physical lake variables. The comparison of the mean depth between 2010 and 2018 (Figure 8) shows a very strong relationship ($R^2 = 0.89$ and *p*-value = $3.619 \times 10^{-6}$), which indicates a good significance of the methodology. Furthermore, the values are located below the bisector, as in no case do we find negative silting (greater lake depth). This leads us to think further that the methodology is correct. Given the small variation in terms of volume in most cases (Figure 7), calculation errors, however accepted, could lead to mathematical anomalies. Mathematical models verify the physical process of sediment accumulation in the lake. The 12 basins analyzed are very different from each other (average surface = $78 \pm 54$ hectares); they mainly have soil covers relating to arable land and forest, so with high variability. We also found evident differences in terms of precipitation. The lakes examined are well distributed and are representative enough of the study area, Tuscany. Some lakes exhibit very high-volume variations (Figure 7). For example, Lake 12964 has a silting rate of 45 tons/year per hectare of basin.

Close to the lake, we found a pig farm on an area of about seven hectares, a practice that has probably greatly affected erosion. Lake 5171 has the largest basin of all the lakes under study, with a greater total length of variability, another important source of sediment when maintenance is limited. Lake 8454 also has a strong silting; in this case, the data available do not show any trend and the lack of knowledge of the specific territory does not allow us to put forward hypotheses. Lake 11525 has a silting rate of 0.21 ton/year per hectare of basin, and has the basin completely covered by forest. The models used provided data compatible with the results of other authors [34,40]. The soil-loss values obtained from the basins show a significant correlation with the silting rate values (Figure 10), regardless of the intrinsic great variability. In some cases, soil loss is greater than the volume variation (8 out of 24, Table 5); in particular, this concerns large basins (about 100 hectares). One reason can be that the sediment produced by the basin does not reach the lake. Another factor that has not been taken into account concerns the suspended sediment lost by the reservoir spillway. Tauro [72] indirectly estimates suspended solids by means of water turbidity. The concentration of suspended solids typically increases with the flow speed. In a lake, the effect of containment and slowdown of the outflows allows greater sedimentation. In fact, the water flowing in the spillways usually appears clear [73]. The opposite behavior occurs in smaller basins (less than 50 hectares), which are characterized by higher silting rates and lower soil loss values. The actual erosion is caused by phenomena that, in some cases, the models are unable to consider, which concern the slopes closest to the lake (such as the case of the lake with pig breeding). Having such present shortcomings in mind, the applied methodology can be useful for building decision-support systems in spatial planning and monitoring areas that show critical characteristics related to hydrological processes [74].

## 5. Conclusions

A new approach for the soil erosion analysis was proposed. It consists of the use of an aquatic boat bringing a GPS sonar. In few minutes, it is able to detect data about water depth in the reservoir. Using an automatic GIS process, it is possible to obtain an estimation of the volume. The methodology was applied to 12 hilly lakes mainly for irrigation purposes, for which the volume (or reservoir capacity) was measured and repeated after 8 years, using comparable instruments (sonar with GPS). The volume variation in this period was compared with soil-loss estimates obtained from well-established models widely known in the scientific community (RUSLE and MMF), obtaining a clear relationship between the two variables. This approach allows low-cost monitoring of the soil erosion phenomena in relation to changes in land use or climate change. Being based on lakes, the analysis can refer to specific portions of the territory. The main advantage is the speed of carrying out the survey on the lake; the instrumentation is inexpensive and it is not necessary to acquire other parameters relating to the basin. The processing procedure can be automated in the GIS environment. The method can be applied to land management issues as a tool for a decision-support system. The harvesting of more data could permit the development of some estimate models about the erosion phenomena based on the silting rate of lakes.

**Author Contributions:** Conceptualization, S.C. and R.G.; methodology, L.G. and S.C.; software, L.B. and S.R.; validation, R.G. and M.I.; formal analysis, Y.G. and L.G.; investigation, Y.G., R.G., S.C.; resources, B.G. and A.O.; data curation, Y.G.; writing—original draft preparation, Y.G.; writing— review and editing, A.O., S.C. and R.G.; visualization, M.I.; supervision, L.B. and A.O.; project administration, B.G.; funding acquisition, B.G. and L.B. All authors have read and agreed to the published version of the manuscript.

**Funding:** This research was carried out with the ordinary funds of the LaMMA Consortium and CNR-IBE and with external funding from the Tuscany Region: Decrees 100/2018 and 1345/2018 "Mappatura della totalità dei laghi in Regione Toscana e costituzione del catasto informatizzato".

**Institutional Review Board Statement:** Not applicable.

**Informed Consent Statement:** Informed consent was obtained from all subjects involved in the study.

**Data Availability Statement:** Not applicable.

**Acknowledgments:** The data used for this article derive from the hilly lakes census of the Tuscany Region.

**Conflicts of Interest:** The authors declare no conflict of interest.

## Appendix A

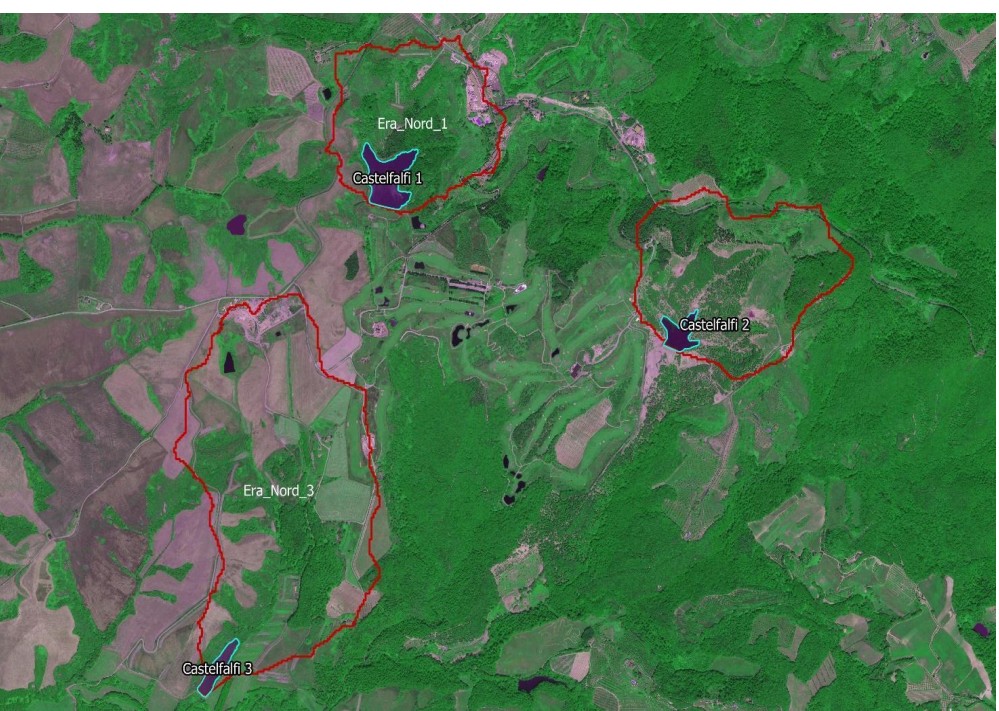

**Figure A1.** Castelfalfi lakes and basins. GID: 8454 = Castelfalfi 1; 12964 = Castelfalfi 2; 8477 = Castelfalfi 3.

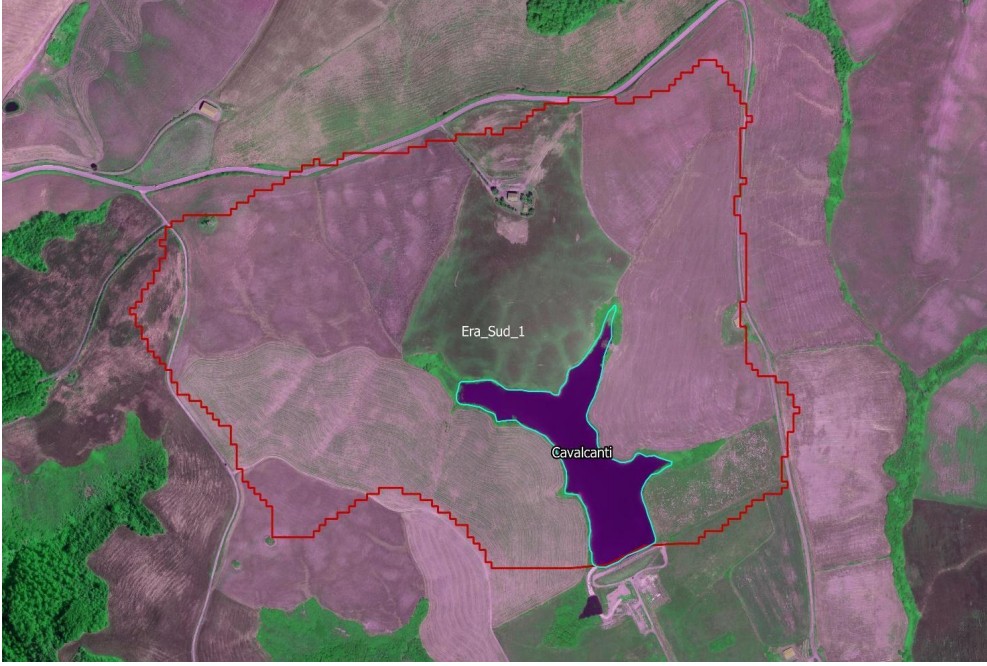

**Figure A2.** Cavalcanti lake and basin. GID: 2629.

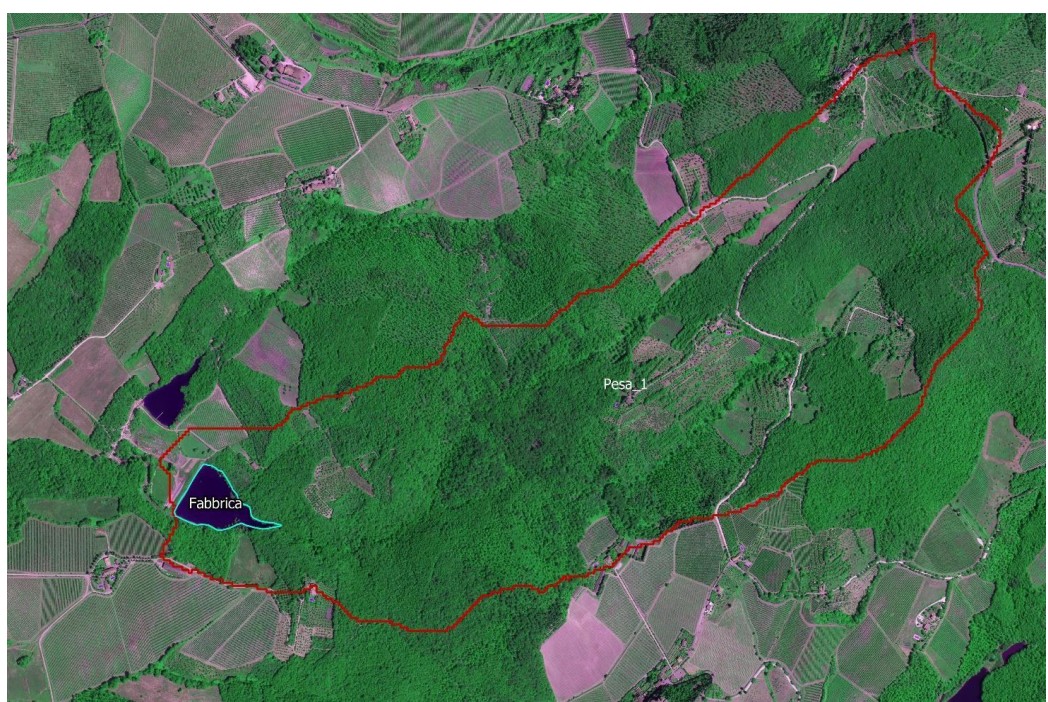

**Figure A3.** Fabbrica lake and basin. GID: 5171.

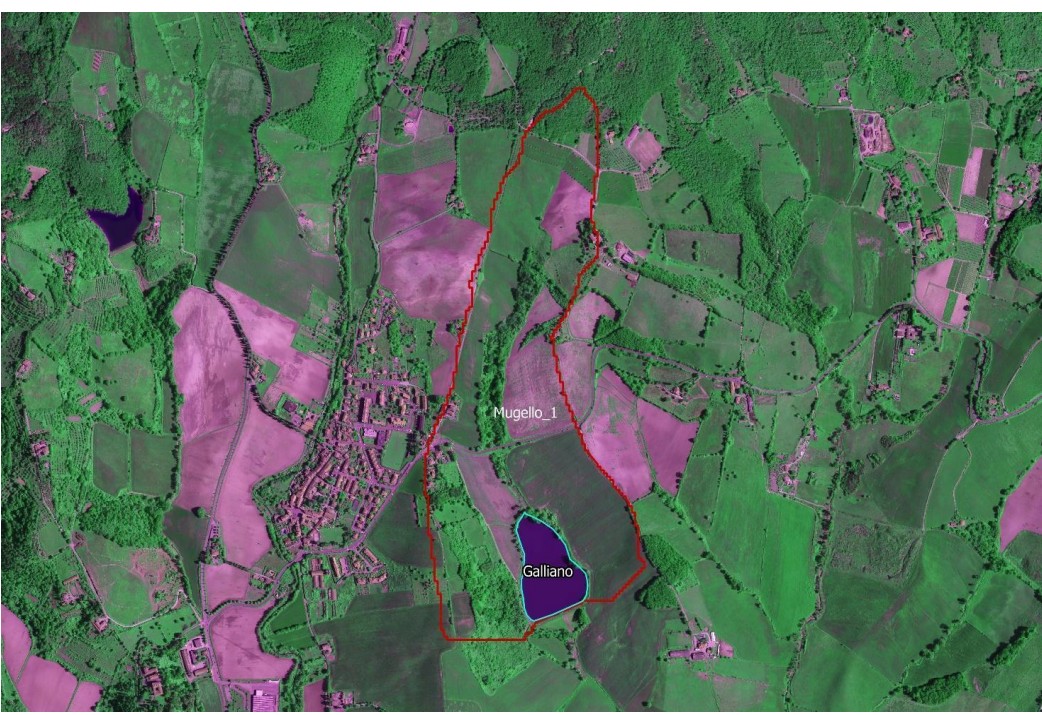

**Figure A4.** Galliano lake and basin. GID: 3036.

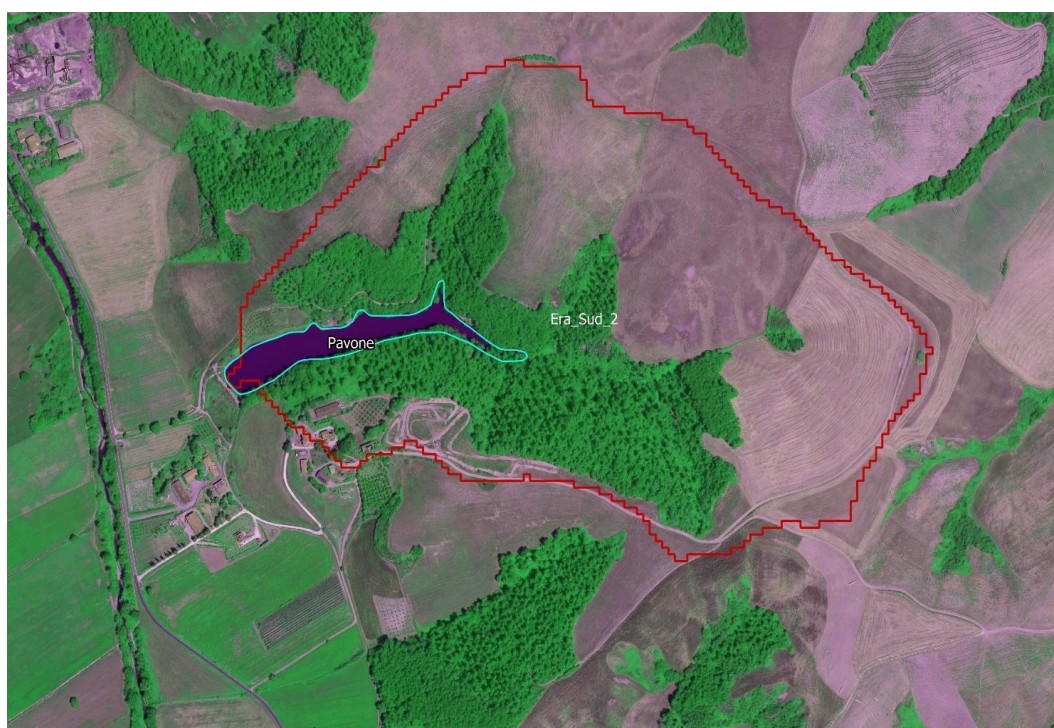

**Figure A5.** Pavone lake and basin. GID: 7438.

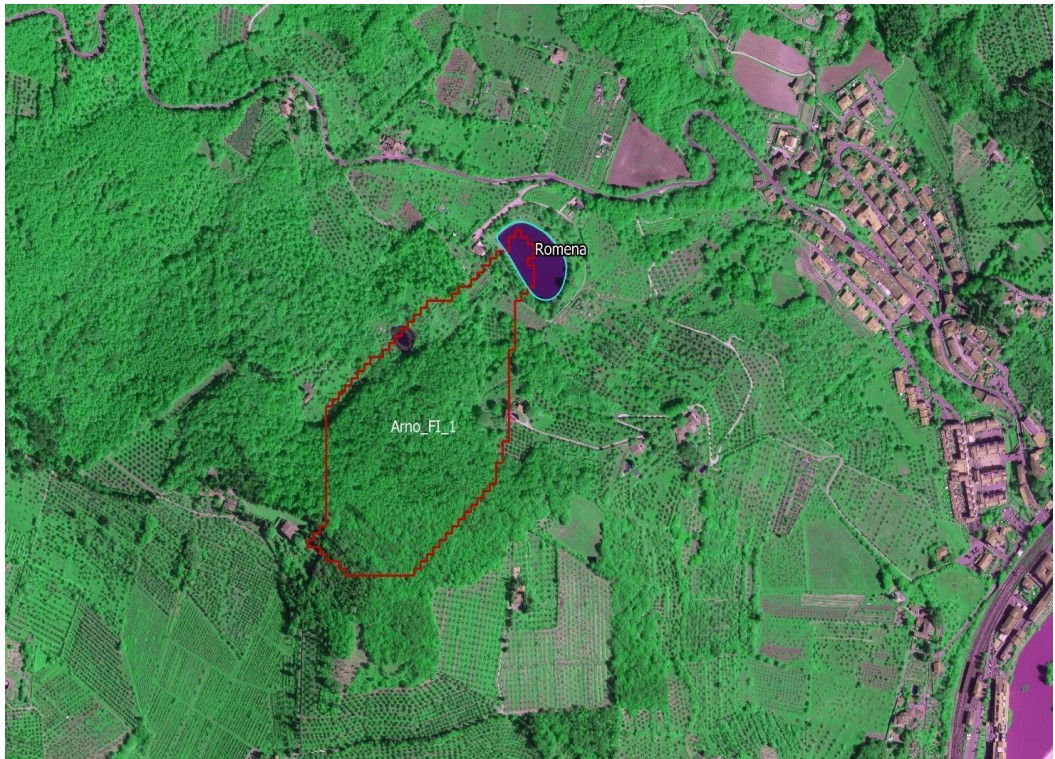

**Figure A6.** Romena lake and basin. GID: 1156.

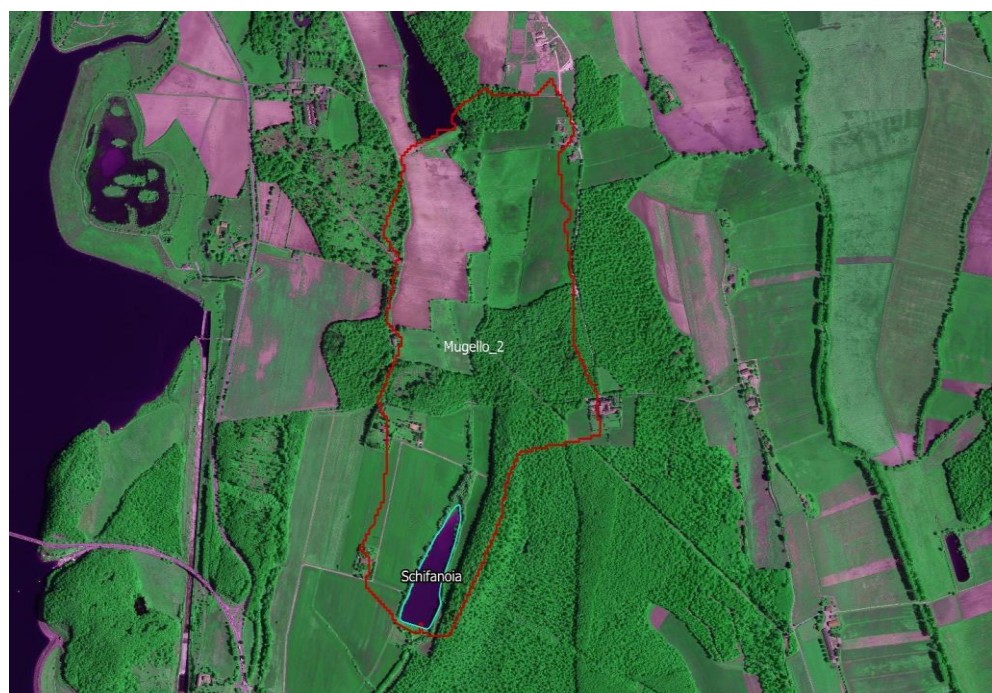

**Figure A7.** Schifanoia lake and basin. GID: 7719.

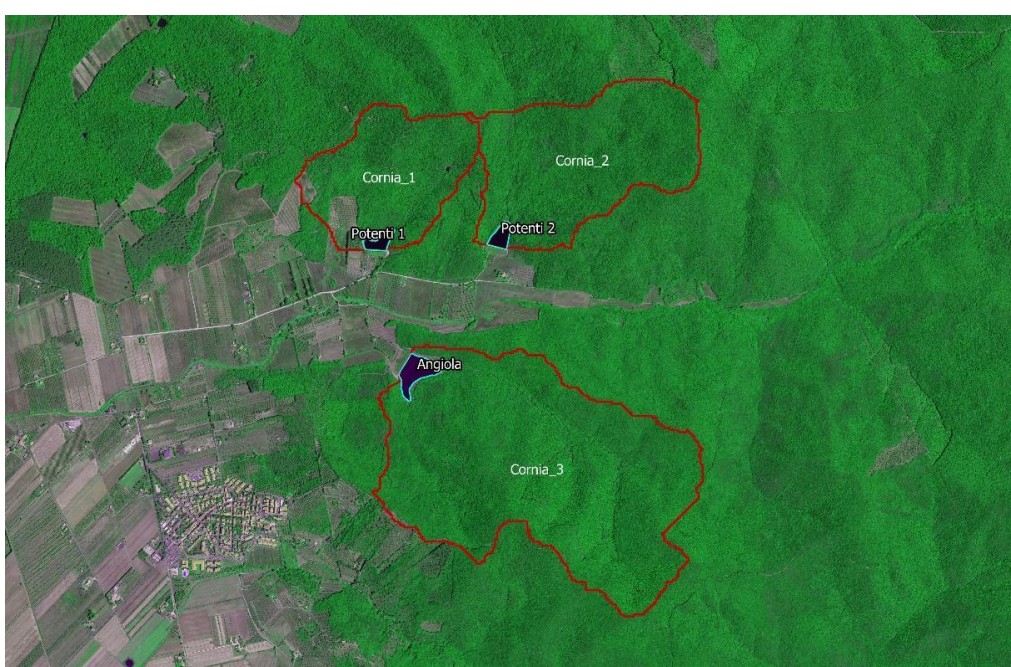

**Figure A8.** Cornia lakes and basins. GID: 8969 = Potenti 1; 8967 = Potenti 2; 11525 = Angiola.

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
