# Peer review of "Assessing Soil Erosion by Monitoring Hilly Lakes Silting"

_sustainability, doi:10.3390/su14095649_

Round 1

Reviewer 1 Report

General remark the article is based on a logical system of comparing field research with the results of analyzes using specialized RUSLE and MMF software. The basic objection to the presented results is related to the use of a digital terrain model with a resolution of 10 m for detailed analyzes. The research should be based on a detailed DTM model, for example Lidar. I suggest using them at work.

Author Response

We fully agree in this observation, unfortunately we do not have usable LiDAR data for the cases examined. Only 2 basins of those analyzed are covered by dem lidar (1x1), carried out over 15 years ago. We have added a sentence to the discussions to state the possibilities for improvement. Thanks.

Reviewer 2 Report

Brief summary

This manuscript describes the use of a novel technique to estimate soil erosion at a basin scale.

Broad comments

The manuscript is worthy, in general is well written with only a few minor errors.

The title is OK.

Keywords: Probably is worthy to add USLE and MMF

Abstract: I think that it should be improved. The methodology and the main results are little explained. 

Tables and figures: Table 2 is not necessary. Some figure captions should be improved (see further comments).  

Some specific comments

L118. Please replace ‘In’ for ‘in’

L121. Please define DTM

L138. Please improved the legend of Figure 4, I do not understand the legend of the land uses. 

L152. Which are the units of volume?

L313. One-sentence paragraph.

L347. Please use Mg y-1.

L369. Please delete ‘the’ before soil.

Author Response

please, look at the attached file, where you can find the answers to the comments. Thanks

Reviewer 3 Report

Review of Assessing soil erosion by monitoring hilly lakes silting

This research compares a remote sensing method of assessing soil erosion via lake siltation to RUSLE and MMF models by comparing two lake volume datasets from 2010 and 2018. The authors assume that catchment sediment production is correlated to reservoir volume loss.

Abstract:

some broad generalizations that soil erosion is assessed through modeling only. This is not correct – many field and lab studies have also been performed on soil erosion.

Line 13 sentence beginning with “Furthermore” implies erosion is the result of direct measurements – the true meaning is unclear.

Introduction:

some English editing is needed throughout.

Several models are described

Materials and Methods:

It is unclear whether the researchers are measuring volume or storage capacity of these lakes. Fig 6 seems to suggest that it is lake volume, but there are many factors that affect water volume: infiltration, precipitation, change in storage capacity…

Please describe briefly the method used to measure the lake volume in 2018, and justify the assumption that the two methodologies are comparable.

Tables appear to be referenced out of order (Line 113-115)

Figure 1 – please label lakes by their name instead of GeoID

Line 127 “the hydrographic was defined basin from a “? Unclear

Figure 3 – is the lake surface the surface area? On the chart legend, use superscript for units of lake surface area. Also please use lake names or shortened names like SCHIF, CAS1, CAS3, POT2, POT1, etc. instead of GeoID

Figure 4 – many questions about this one. Is the y-axis proportion? It reads percent, but then much of the catchment would be unaccounted for. For land use classes (x-asis labels), please use words not codes. Please enlarge figure so that colors can be discerned. Is this figure necessary?

Table 2 and Figure 5, Lines 141-145. The authors indicate that the design volume is unreliable due to unregistered changes during construction, which suggests that these data should be used very carefully. I suggest omitting Table 2 (data are repeated here) and keep Fig 5, but change the symbol for the period 1960-2010 to a dashed line to indicate the uncertainty. Also, use straight lines between each measurement point rather than smoothed lines, and include a symbol on each point to be very clear about the time steps for each measurement period.

In equation 1, what is sigma?

Equation 2, can we really assume that variation in volume is negligible for small changes in surface area?

To this point (line 184), it is unclear how precipitation is factored in. Could delta S be due to climatic factors? How do you rule this out? Is an assumption that increased precipitation leading to increased siltation is irrelevant being made?  This needs clarification.

MMF model – annual precipitation is estimated for each lake from the closest rain gauge. What is the density of rain gauges? This is somewhat imprecise – if lakes and rain gauges are proximal, then some sort of interpolation measure (isohyets, Kriging, IDW, Thiessen polygons) and weighted averaging would give a more accurate estimate of precipitation in each catchment.

Results:

Line 266-269 this description of the data in Figure 9 (correlation matrix) is confusing. Where is “erodibility of lithology” in the figure? Variable names do not appear to match between figure and text. Also for each variable listed, what is the other paired variable? E.g., R = 0.72 between erodibility of lithology and what variable? 

Figure 9 – why is GID included here? Please omit as it is not relevant.

The symbol for correlation is r, not R2

Typo in legend of Figure 8 b

Figure 10 – please use a clustered column graph, the lines imply continuity and are appropriate for a time series, not discrete sites. Please use lake names (or short names) rather than GeoID numbers

Line 304 – what happened with lakes 5171, 8454, and 12964? Should be discussed in the discussion. Is land use different? Different slope? Aspect?

Figure 11 – one should be careful drawing any conclusions from a regression with this few data points. The outlier with silting rate measured over 3000 t/year seems to have a large impact on the MMF regression line.

Discussion:

Line 326-328 – what simplification in the lake profile?  This should be presented in methods too,

Inconsistencies in results for lakes 5171, 8454, and 12964 should be discussed.

This is an interesting paper that could be made more relevant by explicitly showing how findings may be applied. It seems that use of lake volume data to estimate siltation to in turn estimate erosion is labor intensive, and models would be a preferred option, especially if models produce reasonable results.

Similarly, if the aim is to prove that lake surface area change can be used to estimate siltation, I am still unconvinced as the explanation of how precipitation/drought is factored in is not really clear. Promising work, but I feel that the take-away message is not clear and is lost on me.  

Author Response

(The authors gave the same response as above.)

Round 2

Reviewer 3 Report

Please see comments on attached PDF.

Author Response

Thanks again to the reviewer for their attention and care in revising the article.
The point-to-point responses can be found below, or in the PDF in which the review was carried out.
while the changes to the text and images are in the docx, in which the traces of the past revision have been removed.

REV:      analyses

YAMU:       thanks

REV:      repetition

YAMU:       thanks

REV:      please add a scale bar and north arrow. An inset locator map would also be appropriate as this is a global journal.

YAMU:       ok, we added, thanks

REV:      The land cover class codes are meaningless for someone in another country that uses a different land classification scheme YAMU:       (NLCD, for example, in the USA).  
I reiterate a comment from the first review, that these classes should have descriptive names. If unable to add to the figure, there should be an accompanying table or the descriptions should be added to the Figure caption.  This omission interferes with comprehension and is inappropriate in a journal directed to a global audience.

YAMU:       ok, we add the description of the classes in the caption of the image, since the table with the descriptions is inserted further on.

REV:      I see that the authors have changed the lines to dashed lines for the full time period. My comment on the first review was to use dashed lines for the first interval and a solid line for the second interval to indicate the uncertainty about the first data point.
I reiterate the recommendation to use straight lines rather than curved lines. I feel the curved lines are somewhat misleading as they suggest more data points than the three represented in this figure. 

YAMU:       ok, ok, we put dashed lines and solid lines for the second part of the trend. Thanks

REV:      For clarity, I recommend revising to "Among these we find a good correlation between soil loss and the erodibility of ......."

Also, doesn't the red color indicate negative correlation? If that is the case the new text here should read K_lito_Sl, r = -0.72). 

I note a discrepancy between the variable name in Fig 9 (K_lito_Sl) and in this paragraph (KS_lito_Sl). Is there a typo?

YAMU:       yes, sorry they were typos, thanks.

REV:      Could the authors not examine terrain and land use in this watershed to assess whether these may be factors?

YAMU:       we looked at terrein and land use, but found no evidence. As for the lake next to the pig farm, this was verified on the ground, during the surveys, hardly visible via photointerpretation.

REV:      The conclusions indicate that the aim of the research is to estimate soil erosion using repeated bathymetric data from receiving lakes. and that this method is viable and quick and easy. I'm not convinced this field based method is simpler and easier than modeling, especially as model input data are readily available.

YAMU:       to develop the models many surveys and analyzes are necessary, for example regarding the soil, normally the models are applied with data taken from other authors, acquired in the past, often not updated for large areas of territory.
Sudden changes may not be visible, particular land uses, urbanization, climate changes, are phenomena that greatly affect erosion and with the proposed technique we believe that it can be found.

REV:      Here Lakes are referred to by name only, but to compare images with the lake numbers used persistently throughout the paper, they also should be included in all figure captions. 

YAMU:       we have added gids in the captions, so it is easier to cross data with images. thank you